# 3D-FilterMap: A Compact Architecture for Deep Convolutional Neural Networks

**Yingzhen Yang[1], Jianchao Yang[1], Ning Xu[1], Wei Han[2], Nebojsa Jojic[3], Thomas S. Huang[2]**
[1] Snap Research, Venice, CA 90291, USA
superyyzg@gmail.com, {jianchao.yang,ning.xu}@snap.com
[2] Beckman Institute, University of Illinois at Urbana-Champaign
{weihan3,t-huang1}@illinois.edu
[3] Microsoft Research
jojic@microsoft.com

## Abstract

We present a novel and compact architecture for deep Convolutional Neural Networks (CNNs) in this paper, termed 3D-FilterMap Convolutional Neural Networks (3D-FM-CNNs). The convolution layer of 3D-FM-CNN learns a compact representation of the filters, named 3D-FilterMap, instead of a set of independent filters in the conventional convolution layer. The filters are extracted from the 3D-FilterMap as overlapping 3D submatrics with weight sharing among nearby filters, and these filters are convolved with the input to generate the output of the convolution layer for 3D-FM-CNN. Due to the weight sharing scheme, the parameter size of the 3D-FilterMap is much smaller than that of the filters to be learned in the conventional convolution layer when 3D-FilterMap generates the same number of filters. Our work is fundamentally different from the network compression literature that reduces the size of a learned large network in the sense that a small network is directly learned from scratch. Experimental results demonstrate that 3D-FM-CNN enjoys a small parameter space by learning compact 3D-FilterMaps, while achieving performance compared to that of the baseline CNNs which learn the same number of filters as that generated by the corresponding 3D-FilterMap.

## 1 Introduction

With the rise of deep learning, deep Convolutional Neural Networks (CNNs) are popular choices for learning highly semantic and discriminative features for various tasks including image classification. It can be observed that the convolutional layer is important for learning deformation invariant features, e.g. translation invariant features, by convolving the input with filters, and the number of filters influences the quantity of such features. On the other hand, it is widely believed that there is considerable redundancy in the filters, and previous study shows that the parameter space of the filters can be significantly reduced by various model compression techniques such as quantization or exploiting the low-rank and sparse representation of the filters Han et al. (2015); Ioannou et al. (2016); Yu et al. (2017)

Based on the observation of the redundancy in the filters, the above discussion leads us to an interesting question: is there a way of generating the filters from a reduced parameter space? If it is achievable, we then have a solution that learns a compact CNN from scratch. We propose 3D-FilterMap as a novel and compact structure of organizing the filters to achieve this goal. 3D-FilterMap is a 3D matrix from which the filters are extracted from as overlapping 3D submatrices (see Figure 1). When a certain number of filters are extracted from a 3D-FilterMap wherein nearby filters share weights, the parameter space of the 3D-FilterMap is much smaller than that of the same number of independent filters to be learned in the conventional convolution layer. In this manner, 3D-FilterMap enables a way of directly learning a compact CNN. In contrast, the model compression literature broadly adopts a two-step approach: learning a large CNN first, then compressing the model by various model compression techniques such as pruning, quantization and coding Han et al. (2015); Luo et al.

The work of Yingzhen Yang was supported in part by an IBM gift research grant to the University of Illinois at Urbana-Champaign.

(2017), or utilizing the low-rank or sparse representation of the filters based on the redundancy in them Ioannou et al. (2016); Yu et al. (2017).

## 2 FORMULATION

---

Description of Forward and Backward Operation in a convolution layer of 3D-FM-CNN with 3D-FilterMap

---

**Input:**

**Forward:** Extract $K$ overlapping filters $\{\mathbf{F}^{(k)}\}_{k=1}^{K}$ from the 3D-FilterMap $\mathbf{F}^{(M)}$, and each filter $\mathbf{F}^{(k)} \in \mathbb{R}^{S_1 \times S_2 \times C}$. Then convolve these $K$ filters with the input of the convolution layer, where $K$ is the number of filters in the corresponding convolution layer of the baseline CNN.

**Backward:** First obtain the gradient of all the $K$ filters as $\{\mathbf{G}^{(k)}\}_{k=1}^{K}$ where each $\mathbf{G}^{(k)} \in \mathbb{R}^{S_1 \times S_2 \times C}$. For each element $j \in \mathbf{F}^{(M)}$, its gradient is computed by

$$j' = \frac{\sum\limits_{k=1}^{K} \sum\limits_{t} \mathbf{G}_t^{(k)} \mathbb{I}_{\mathbf{T}(t)=j}}{\sum\limits_{k=1}^{K} \sum\limits_{t} \mathbb{I}_{\mathbf{T}(t)=j}} \qquad (1)$$

---

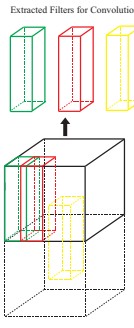

Extracted Filters for Convolution

3D-FilterMap

**Figure 1:** Illustration of a 3D-FilterMap. The three filters in green, red and yellow are three overlapping filters extracted from the 3D-FilterMap.

The idea of 3D-FilterMap is inspired by epitome Jojic et al. (2003), which is developed in the computer vision and machine learning literature for learning a condensed version of Gaussian Mixture Models (GMMs). In epitome, the Gaussian means are represented by a two dimensional matrix wherein each window in this matrix contains parameters of the Gaussian means for a Gaussian component. The same structure is adopted for representing the Gaussian covariances.

The above characteristics of epitome encourages us to arrange filters in a way similar to epitome in the proposed 3D-FilterMap Convolutional Neural Networks (3D-FM-CNN). More concretely, each convolution layer of 3D-FM-CNN has a 3D matrix named 3D-FilterMap, wherein the overlapping 3D submatrices in the 3D-FilterMap play the same role as the filters in the conventional convolution layer of ordinary CNN. 3D-FM-CNN and its baseline CNN have the same architecture except that each convolution layer of 3D-FM-CNN comprises a 3D-FilterMap rather than a set of independent filters. The 3D-FilterMap in each convolution layer of 3D-FM-CNN generates the same number of filters as the corresponding convolution layer in the baseline CNN. Instead of learning a set of independent filters for each convolution layer of CNN, a compact 3D-FilterMap of much smaller parameter size is learned for each layer of 3D-FM-CNN.

An illustration of 3D-FilterMap is shown in Figure 1 which illustrates an example of 3D-FilterMap and how overlapping filters are extracted from it. Suppose that a convolution layer of the baseline CNN model has $64$ filters of channel $64$ and spatial size $3 \times 3$, the corresponding convolution layer in the 3D-FM-CNN has a 3D-FilterMap of size $64 \times 8 \times 8$. The $64$ $64 \times 3 \times 3$ filters are sampled by striding along each spatial dimension by 2, and striding along the dimension of the channel by 16. The ratio of the parameter size of the $64$ independent filters to that of the corresponding 3D-FilterMap is $\frac{64 \times 64 \times 3 \times 3}{64 \times 8 \times 8} = 9$, indicating that the parameter space of the 3D-FilterMap is much smaller than the independent filters in the baseline CNN.

Formally, suppose a 3D-FilterMap should generate $K = K_1 \times K_2 \times K_3$ filters of size $S_1 \times S_2 \times C$ where $(S_1, S_2)$ is the spatial size of filter and $C$ is the channel size. Let the filter sampling stride along two spatial dimensions of the 3D-FilterMap are $x$ and $y$, and the sampling stride along the channel dimension of the 3D-FilterMap is $z$. Then the dimension of the 3D-FilterMap is $(K_1 x, K_2 y, K_3 z)$, where $(K_1 x, K_2 y)$ is the spatial size and $K_3 z$ is the channel size. In this paper we set the channel size of the 3D-FilterMap to be $K_3 z = C$, which is based on our observation that the weights along the channel can be shared more frequently without hurting the performance. Therefore, the ratio of the parameter size of the $K$ independent filters to that of the corresponding 3D-FilterMap is $\text{ParamRatio} = \frac{K \cdot S_1 \cdot S_2 \cdot C}{K_1 x \cdot K_2 y \cdot K_3 z} = \frac{K_1 K_2 K_3 \cdot S_1 \cdot S_2 \cdot C}{K_1 x \cdot K_2 y \cdot K_3 z} = \frac{S_1 \cdot S_2 \cdot C}{x \cdot y \cdot z} = \frac{S_1 \cdot S_2}{x \cdot y} \cdot K_3$. In a typical setting where the spatial stride is smaller than the corresponding filter size, i.e. $x < S_1$, $y < S_2$, the 3D-FilterMap has a compact size. Also note that a larger $K_3$, namely the sampling number along

the channel dimension, leads to a more compact 3D-FilterMap in the manner that the weights of the 3D-FilterMap along the channel dimension are shared more frequently by nearby filters.

Algorithm 1 describes the forward and backward operation in a convolution layer of 3D-FM-CNN with 3D-FilterMap. We use the mapping $\mathbf{T}$ which maps the indices of the elements of the extracted filters to the indices of the corresponding element in the 3D-FilterMap. Namely, for a filter $\mathbf{F}^{(k)}$ and the 3D-FilterMap $\mathbf{F}^{(M)}$, $\mathbf{F}_t^{(k)} = \mathbf{F}_{\mathbf{T}(t)}^{(M)}$ (please refer to the notations in Algorithm 1). The mapping $\mathbf{T}$ is used to conveniently track the origin of the elements of the filters extracted from the 3D-FilterMap.

## 3 EXPERIMENTAL RESULTS

We demonstrate the performance of 3D-FM-CNN in this section by the comparative results between 3D-FM-CNN and its baseline CNNs. Using ResNet He et al. (2016) and DenseNet Huang et al. (2017) as the baseline CNNs, we design 3D-FM-CNN by replacing each $3 \times 3$ convolution layer of ResNet or DenseNet by a convolution layer with 3D-FilterMap in the 3D-FM-CNN. We train 3D-FM-CNN and the baseline CNNs on the CIFAR-10 data, and show the testing accuracy and the parameter number of all the models in Table 2. ResNet-18-FM indicates the 3D-FM-CNN using ResNet-18 as the baseline CNN, and similar naming convention is used for other models. The convolution layers of ResNet and DenseNet have filters of spatial size of either $3 \times 3$ or $1 \times 1$. We replace all the $3 \times 3$ convolution layers by the convolution layers with 3D-FilterMap in the corresponding 3D-FM-CNN so as to demonstrate the capability of 3D-FilterMap to represent the filters that capture the spatial pattern in the input. Note that we do not use 3D-FilterMap to represent $1 \times 1$ convolution layers since $1 \times 1$ convolution layer is primarily used for dimension reduction. We design the size of the 3D-FilterMap according to the number of filters in the corresponding convolution layer of the baseline CNN. Throughout this section, we set the size of the 3D-FilterMap according to Table 1, and use the spatial stride $x = y = 2$. We do not specifically tune $(K_1, K_2, K_3)$, and one can surely choose other settings for these three hyperparameters as long as their product matches the number of filters in the baseline CNN.

It can be observed in Table 2 that 3D-FM-CNN achieves comparable performance with a compact parameter space for different baseline CNNs including ResNet-18, ResNet-34, ResNet-50 and DenseNet-121. DenseNet-121 indicates DenseNet with a growth rate of 32 and 121 layers. Since most of the parameters of ResNet-18 and ResNet-34 are in the $3 \times 3$ convolution layers, 3D-FM-CNN exhibits a relatively high parameter ratio (ratio of the parameter number of the baseline CNN to that of the 3D-FM-CNN) of around 11.7. ResNet-50 and DenseNet-121 have a significant amount of parameters in $1 \times 1$ convolution layers, therefore, the parameter ratio is not that high. However, in the latter case 3D-FM-CNN slightly generalizes better evidenced by an even better testing accuracy.

Table 1: The Size of 3D-FM-CNN

| #Filters | $K_1$ | $K_2$ | $K_3$ |
|---|---|---|---|
| 12 | 2 | 3 | 2 |
| 32 | 4 | 4 | 2 |
| 64 | 4 | 4 | 4 |
| 128 | 8 | 4 | 4 |
| 256 | 8 | 8 | 4 |
| 512 | 8 | 8 | 8 |

Table 2: Performance of 3D-FM-CNN and the corresponding baseline CNN

| Model | ResNet-18 | ResNet-18-FM | ResNet-34 | ResNet-34-FM | ResNet-50 | ResNet-50-FM | DenseNet-121 | DenseNet-121-FM |
|---|---|---|---|---|---|---|---|---|
| Accuracy | 94.18% | 93.55% | 94.72% | 94.25% | 95.16% | 95.47% | 95.13% | 95.45% |
| #Parameter | 11.2M | 0.95M | 21.3M | 1.8M | 23.5M | 13.1M | 7.0M | 5.3M |

## 4 CONCLUTION

We present 3D-FilterMap Convolutional Neural Networks (3D-FM-CNNs) as a compact architecture for deep convolutional neural networks in this paper. In contrast with learning a set of independent

filters in the conventional convolution layer, the convolution layer of 3D-FM-CNN learns a compact 3D-FilterMap. The acutal filters are extracted from the 3D-FilterMap in a way such that nearby filters share weights. Thanks to the weight sharing scheme, 3D-FM-CNN has a much smaller parameter space than its baseline CNN when it generates the same number of filters as the baseline CNN. Experimental results demonstrate the effectiveness of 3D-FM-CNN in learning a compact model while exhibiting performance comparable to the baseline CNNs.

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
