# OpenReview forum: "3D-FilterMap: A Compact Architecture for Deep Convolutional Neural Networks"
_ICLR.cc/2018/Workshop — Accept_

### Official Review · AnonReviewer3 · 2018-03-07

**Rating:** 6
**Confidence:** 4

**Review:**

This paper proposes a filter sharing methods for a convolutional layer in neural networks. Instead of learning several independent filters, the proposed method uses a larger 3D tensor as the parameter of the convolutional layer. Sliding window scheme is used along the 3 dimensions to crop sub-tensors, and each tensor is taken as a convolutional filter.

Pros:

The idea of geometric filter sharing is interesting. It provides a simple way to reduce the number of parameters without changing the network architecture and training strategy.

The experimental results show that the proposed method can reduce the number of parameters significantly with a minor performance loss.

Cons:

The number of output channels for a convolutional layer is not arbitrary in the proposed methods. It must be the multiplication of the numbers of cells in all 3 axes.

The proposed method cannot reduce the computational complexity. Given the existence way to shrink both the number of parameters and computation cost, it is not very clear how significant the proposed method is in practice,

The paper made interesting observations, but it will be great to have more insights on why sharing the filters in the proposed way is reasonable.

Presentations:
1. The abstract is not easy to follow. Several undefined terminologies pop up, and the main idea of the paper is not clearly highlighted.
2. It seems more common to say 3D tensor rather than 3D matrices.

Overall, the paper is interesting, and it has the potential to be extended to a long paper.

---

### Official Review · AnonReviewer2 · 2018-03-09
**Good paper, novel approach**

**Rating:** 7
**Confidence:** 3

**Review:**

This paper presents the approach of 3D-FilterMap, an alternative architecture to standard CNNs where the weights of the learned filters are shared locally instead of being independent. Experiments on CIFAR-10 are presented and results show that the proposed approach yields comparable performance with standard architectures with less parameters.

Pros:
- The main idea of the paper is very interesting. Learning compact shared filters in the CNN instead of independent ones is a novel and substantial contribution, which could have an impact on the current efforts to reduce neural network size.
- The experiments are convincing, the results are promising and validate the approach.

Cons:
- The paper is sometimes not easy to follow, with some clarity and formatting issues.

Minor comments:
- The second paragraph of Section 2 uses the word "3D-FF-CNN" seven times in a five sentences paragraph, the authors should improve the writing.
- The equations embedded in the text in Section 2 make it difficult to follow.
- The dataset used in Section 3 should be more clearly written (maybe in a small paragraph).

Overall, this paper presents a novel and promising approach for learning compact CNNs, I recommend acceptance.

I am looking forward to a full version of this work, with experiments on larger dataset and specially an analysis on the filters: it should be interesting to compared the learned filters between the proposed approach and the standard CNNs.

---

### Decision · Program_Chairs · 2018-03-20
**ICLR 2018 Workshop Acceptance Decision**

**Decision:**

Accept

**Comment:**

Congratulations, your paper was accepted to the ICLR workshop.